# Aircraft Detection Using Phase-Sensitive Optical-Fiber OTDR

**DOI:** 10.3390/s21155094

**Published:** 2021-07-28

**Authors:** Yunpeng Cai, Jihui Ma, Wenfa Yan, Wenyi Zhang, Yuhang An

**Affiliations:** Key Laboratory of Transport Industry of Big Data Application Technologies for Comprehensive Transport, Ministry of Transport, Beijing Jiaotong University, Beijing 100044, China; cyp@bjtu.edu.cn (Y.C.); yanwenfa@bjtu.edu.cn (W.Y.); wyzhang@bjtu.edu.cn (W.Z.); 19120785@bjtu.edu.cn (Y.A.)

**Keywords:** aircraft detection, Φ-OTDR, seismic vibration signal, doppler effect

## Abstract

Aircraft detection plays a vital role in aviation management and safe operation in the aviation system. Phase-Sensitive Optical Time Domain Reflectometry (Φ-OTDR) technology is a prevailing sensing method in geophysics research, structure inspection, transportation detection, etc. Compared with existing video- or radio-based detection methods, Φ-OTDR is cost-effective, suitable for long-distance detection, and resistant to severe weather conditions. We present a detection system using Φ-OTDR technology and analyze the character of the acoustic signal of aircraft. Instead of runway monitoring in the airport or noise detection in the air, this study focuses on the detection of seismic vibration signal excited by the sound of aircraft. The Chebyshev filter is adopted to eliminate the impact of background noise and random noise from the original vibration signal; the short-time Fourier transform is used for time-frequency analysis. The experimental results showed that the seismic vibration signal excited by the aircraft sound is mainly low-frequency, which is under 5 Hz. Time delay of aircraft vibration signal in different locations of the optic fiber is recorded by the sensing system. The Doppler effect is also revealed by the time-domain analysis: the frequency increases when the aircraft is approaching and decreases when the aircraft moves away.

## 1. Introduction

With the development of the airline industry, airplane detection is an essential issue, providing the foundation for aviation management. The acoustic signal generated by airplanes has been deeply investigated in past decades, which is of great importance for managing urban noise pollution [1,2]. However, this type of signal is also widely accepted for aircraft monitoring, recorded by acoustic sensors in many studies [3,4,5,6]. Eibl [7] used seismometer recordings to locate and track helicopters. Meng [8] analyzed the characteristics of airplanes recorded by a dense seismic array near Anaza, California. These works demonstrate that the acoustic signal-based sensing method may be an effective supplement to existing video- or radio-based systems.

Compared with traditional acoustic sensing methods like seismometers, distributed fiber-optic acoustic sensing (DAS) is suitable for long-distance detection and is resistant to severe weather conditions. One of the most mature technologies of DAS is optical time domain reflectometry (OTDR) [9], which uses optical fibers as signal sensing units and data transfer components. Φ-OTDR is a technical solution that uses the phase change of backscattered light to restore the vibration signal. This type of system can yield a sum of backscattered intensities from each scattering center, which allows revealing simulations and signals outside fiber optic cables, such as stress and vibration. Analysis of the vibration signals induced by moving objects makes it possible to monitor vehicles, railways, planes, and other transportation [10].

Discussions regarding the detection of vehicles, trains, and subways using Φ-OTDR have dominated research in recent years. Dou and Lindsey [11] used DAS to detect vehicles by “hearing” the traffic noise in Stanford. Liu [12,13] used phase-sensitive OTDR to estimate the speed of vehicles on campuses and mines, respectively. Xu [14] presented a dual-channel Φ-OTDR (DC-Φ-OTDR), including two single-channel Φ-OTDRs (SC-Φ-OTDR), to locate running vehicles on a highway. Peng [15] used phase-sensitive OTDR to realize long-distance and dynamic speed estimation and location of trains. Timofeev et al. [16,17] proposed a railway detecting system based on coherent-optical time domain reflectometry (C-OTDR) to determine the location, mass, and speed of a train. Kepak et al. [18] presented a fiber-optic detection system (FODS) utilizing Mach–Zehnder fiber-optic interferometry to monitor train velocity in the Prague subway system.

Scholars also have conducted a series of studies on aircraft detection using fiber-optic sensors (FOS), especially for aircraft health monitoring. FOSs for monitoring strain in aircraft structures can be classified into the following categories [19]: backscattering-based FOS [20], interferometric FOSs [21], and grating-based sensors (FBG). Among these different technologies, grating-based sensors (FBG) are the most mature and widely employed optical sensors for structural health monitoring due to their high spatial resolution and stress sensitivity [22,23]. However, the backscattering-based (DAS) FOS have the advantages of distributed sensing and long-distance monitoring, which are very suitable for aircraft operation monitoring. Borinski [24] demonstrated the application of extrinsic Fabry–Perot interferometric (EFPI) technology in the measurement of pressure, acceleration, and skin friction on high-performance aircraft. Carcia Merlo [19] proposed ground-based distributed optical fiber sensors to detect the position and movement of airplanes. Bakhoum [25] used the distributed acoustic sensing for the monitoring and measurement of airplane flutter. These works demonstrated the feasibility of DAS to detect the airplane. Compared with existing airplane detecting methods such as radar, DAS has the advantages of long-distance, high-sensitivity, and cost-effectiveness. Additionally, optical fiber is a passive sensor and independent of any electromagnetic signal. As a result, DAS can be adapted to harsh environments and has anti-inference to severe weather conditions [26].

While most existing studies engage with airplane operation acoustic signal sensing focus on runway monitoring in the airport or noise detection in the air, few works involve detecting seismic waves excited by the sound of airplanes. When an aircraft is flying at a low altitude, it produces a loud sound that spreads to the ground, exciting seismic waves to converse and propagate on the ground. The study of this signal’s characteristics has practical implications for the detection of aircraft off-airport. This paper proposes an airplane detecting system that uses the Φ-OTDR system to discover the character of seismic waves excited by the sound of airplanes. A field experiment was designed to collect seismic waves data in Qinhuangdao from 12 October to 13 October 2020. The results demonstrated that high-frequency components are lost in the conversion process, and seismic waves recorded by the Φ-OTDR system are mainly low-frequency signals under 5 Hz. The Doppler effect was also shown by a time-frequency analysis method using the short-time Fourier transform.

The organization of this paper is as follows: Section 2 introduces the acoustic signal of aircraft and the experimental setup. The results are provided in Section 3. Analysis and discussion are provided in Section 4. Finally, Section 5concludes the results of this work and discusses limitations and problems to be solved in the future.

## 2. Materials and Methods

### 2.1. Seismic Waves Caused by Airplane Noise

Aircraft noise is the sum of sound radiation from various noise sources from the aircraft during the flight. The elasticity and inertia of air allow it to act as a medium for aircraft noise transmission. Vibrations that occur in certain parts of aircraft are transmitted to the surrounding medium and propagate outward from the noise source [1].

When the sound waves arrive on the ground, some of them access the surface from the air, becoming seismic waves through the processes of transformation and excitation. This seismic wave signal implies the flight characteristics of the aircraft and could be an effective tool for aircraft detection. For the experiment described in this paper, we propose an Φ-OTDR sensing system to record the seismic vibration signal.

### 2.2. Phase-Sensitive Optical-Fiber OTDR

For a typical Φ-OTDR system, as shown in Figure 1, the vibration signal causes phase changes of backscattering light at locations detected by the photosensitive device [25]. The locations of the vibration signals are determined by the return time of the backscattering light.

### 2.3. Field Experiment Environment

The field experiment was conducted near an airport in Qinhuangdao using the city’s underground fiber-optic cable between two information centers as the sensing unit. The fiber-optic cable was buried in the subsurface at a depth of 1–1.5 m and the length used in this experiment was 8.1 km. The experimental scenario is shown in Figure 2. The satellite map of the experiment area is shown in Figure 3.

For the Φ-OTDR system used in this experiment, the time interval of collecting data samples at each acquisition point was 1 ms. The spatial resolution was 10 m (meaning that every 10 m of the optic fiber can be regarded as a separate sensor), and the spatial sampling interval was 4 m so that the whole optic fiber could be divided into different channels, each measuring 4 m. The length of the optical fiber cable used in the test was 8.1 km, and the Φ-OTDR system recorded 2025 channels. Figure 4 illustrates the Φ-OTDR system and fiber-optic cable distribution frame used in the experiment.

The hardware of the DAS system is made by Key Laboratories of Transducer Technology, Institute of Semiconductors, Chinese Academy of Sciences, Beijing. It mainly includes a laser, acousto-optic modulator (AOM), Erbium-doped fiber amplifier (EDFA), circulator, fiber Bragg grating (FBG), photoelectric detector, data acquisition, and a signal processing unit. The power of the DAS system is about 20 W and it can operate for over 40 h on an outdoor power supply.

## 3. Results

### 3.1. Time-Domain Analysis

The data supporting this study was collected from 12 October to 13 October 2020. The original signal was recorded as the aircraft flew over the fiber-optic. The sound produced by the aircraft propagates to the ground, which in turn makes occasional contact with the surface and is recorded by the fiber-optic buried in the shallow ground. There was a lot of background noise on the urban surface, such as noise caused by moving vehicles and mechanical operations. We selected 700 channels that are well-recorded to display the sound produced by the aircraft.

The strain on the fiber changes the refractive index of the light wave, resulting in a change to the phase of the Rayleigh scattered light, which in turn is detected by the optical demodulator. The physical quantity detected by the DAS system is the strain, which can be used to reflect the characteristics of the vibration signal, such as amplitude, phase, frequency, etc. The amplitude of the signal in the Figures is the phase change caused by the strain. In this study, the peak of absolute value for the signal range was from 3 × 10^3^ to 4 × 10^3^ rad/sqrt (Hz). The DAS system used in this experiment presented a powerful signal acquisition ability with an average system noise of 4.79 × 10^−4^ rad/✓Hz, and a minimum detectable strain of 10.4 pε/✓Hz. Since the fiber optic sensor is a single component sensor, the direction of the detected strain is mainly along the fiber laying direction, which is approximately perpendicular to the flight direction of the aircraft.

As shown in Figure 5, aircraft exude periodic roars as they fly over a city. The fiber-optic cable lies in the north-south direction, and the aircraft fly in the east-west direction. The signal strength reaches its maximum when the aircraft reaches above the fiber-optic cable and gradually decreases as the aircraft moves away.

To release real-time airplane location or speed monitoring, it is important to record the signal time delay in different locations of the optic fiber.

Figure 6a shows the aircraft noise arriving at both ends of the observation fiber with a time delay of 0.012 s. Figure 6b shows the aircraft noise arriving at both ends of the observation fiber with a time delay of 0.018 s.

### 3.2. Frequency-Domain Analysis and Signal Denoising

Three aircraft were selected as a sample to analyze the frequency domain of the signal. The frequency distributions of the signals are shown in Figure 7.

As shown in Figure 7, the seismic vibration signals of all three aircraft excited by the sound waves are mainly low-frequency signals under 5 Hz.

According to the frequency distributions of the signal, a Chebyshev low-pass filter was designed to eliminate background noise and other random noise. The Chebyshev filter amplitude versus frequency is shown in Equation (1).
(1)$$G_{n}\left(\omega \right)\;=\;\left|H_{n}\left(j\omega \right)\right|\;=\;\frac{1}{\sqrt{1+\epsilon^{2}T_{n}{}^{2}\left(\frac{\omega }{\omega_{0}}\right)}}$$
where |ϵ| < 1, $\left|H_{n}\left(j\omega \right)\right|\;=\;\frac{1}{\sqrt{1+\epsilon^{2}}}$ is the magnification of the filter at the cut-off frequency $\omega_{0},\;T_{n}\left(\frac{\omega }{\omega_{0}}\right)$ represent Chebyshev polynomials. The pass-band cut-off frequency is 5 Hz and the in-band attenuation is 0.1 dB. While the stop-band cut-off frequency is 500 Hz, the stop-band attenuation is 300 dB. The signal before denoising and after denoising is shown in Figure 8.

In approximately channel 300, there is a strong noise induced by vehicles that with a similar frequency distribution to the airplane signal. Therefore, it cannot be eliminated from the original signal by the Chebyshev lowpass filter. We obtained a good denoising result in other channels, as shown in Figure 8c. The SNR is defined as a voltage ratio between the peak of the disturbing signals (S) and the fluctuant range of the noises (N) with the equation of 20 lg(S/N). The SNR of this signal before filtering was 25.9 dB and raised to 48.8 dB for the proposed method.

### 3.3. Time-Frequency Analysis

The short-time Fourier transform was adopted to analyze the time-frequency features. As shown in Figure 9, the frequency does not remain unchanged during that time; the Doppler effect of the airplane signal is revealed: the frequency increases when the aircraft is approaching and decreases when the aircraft is moving away.

## 4. Discussion

Φ-OTDR is a type of distributed sensing technology that is suitable for long-distance detection. However, it is a single-component sensing technology that fails to distinguish the direction of the signal. Using sensory fiber-optic cables laid in multiple directions (e.g., criss-cross) instead of a single direction may be one solution. Moreover, when the aircraft is in the air, detection based on distributed acoustic sensing (DAS) is a big challenge as the signal has a large radiation range, which means different channels in the radiation range record the signal almost simultaneously; this probably leads to operation detection failure in real-time location and speed detection. Therefore, the length of the sensory fiber-optic cables should be far longer than the radiation range of the aircraft signal direct wave to ensure that the signal reaches different locations on the fiber optic cable with a more significant time delay.

From existing works regarding acoustic signal detection of airplanes, the frequency range of the signal can be concluded: the frequency of vibration signals generated by aircraft on the runway can range from about 20 Hz to 200 Hz [7,8], and the noise frequency generated by the aircraft mainly concentrates from 250 Hz to 3500 Hz [2]. However, the seismic signal recorded in this study is low-frequency signal under 5 Hz. Bakhoum demonstrated that the frequency of airplane flutter is from 5 Hz to 8 Hz in his work [26]. The reason why only a low-frequency signal was recorded by the Φ-OTDR system is difficult to prove, but these are two hypotheses First, the high-frequency components are lost during the excitation and conversion process from sound signal to seismic vibration signal; second, the frequency of the acoustic signal changes significantly when it accesses the ground from the air. The excitation and conversion process mechanisms should be highlighted in future research.

## 5. Conclusions

This paper describes a novel fiber-optic phase-sensitive Φ-OTDR under study and development for its use in the continuous monitoring of airplanes off-airport. The system is designed to operate over 8.1 km-long underground optical fiber sensors and employ low-noise solid-state lasers. The signal was analyzed both in the time domain and frequency domain. The Chebyshev filter was adopted to eliminate the impact of background noise and other random noise. Additionally, the signal time-frequency features were analyzed using the short-time Fourier transform. The experimental results show that the seismic wave excited by the aircraft sound wave recorded by the sensing system is mainly low-frequency which is under 5 Hz. The reason why only a low-frequency signal was recorded by the Φ-OTDR system is provided in the discussion section. The Doppler effect was revealed by the time-frequency analysis result. The paper also provides suggestions on how to conduct effective testing on an aircraft’s real-time location and speed detection.

The results demonstrate that the Φ-OTDR is a promising airplane sensing method. This study is an exploratory attempt in this new area, and merits many works in the future: Firstly, the excitation and conversion mechanisms of low-frequency signals must be further investigated; secondly, better experimental solutions should be designed to realize aircraft operation detection; thirdly, further research is needed on the algorithms of seismic wave signal-based airplane detection, such as speed estimation and aircraft location.

## Figures and Tables

**Figure 1 sensors-21-05094-f001:**
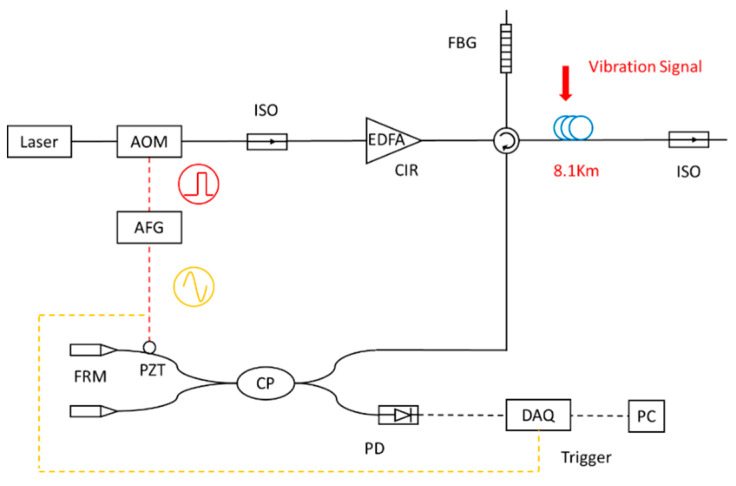
Setup of the Φ-OTDR system.

**Figure 2 sensors-21-05094-f002:**
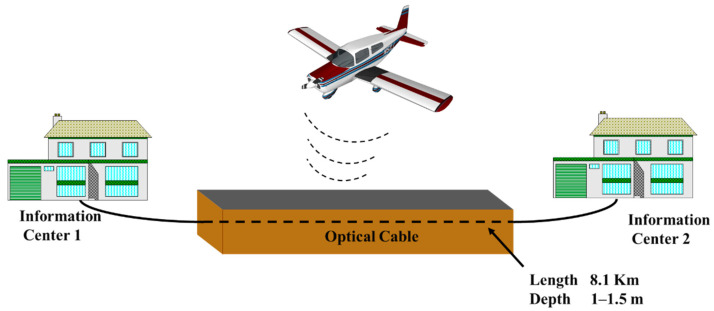
The experiment scenario.

**Figure 3 sensors-21-05094-f003:**
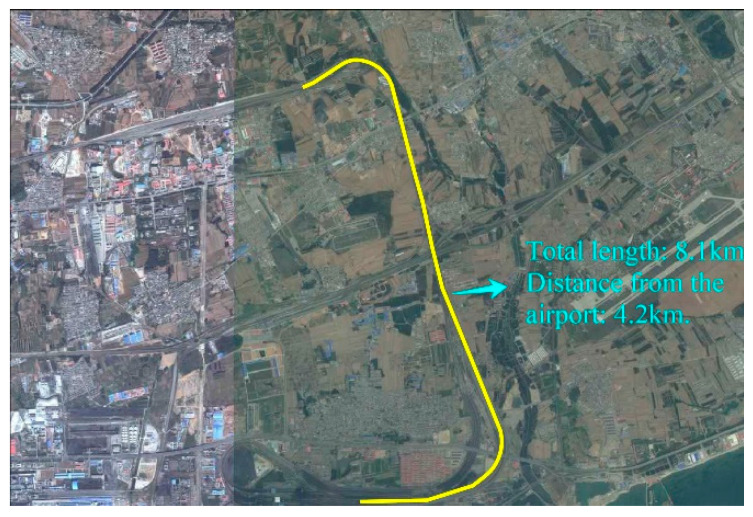
Satellite map of the experiment area.

**Figure 4 sensors-21-05094-f004:**
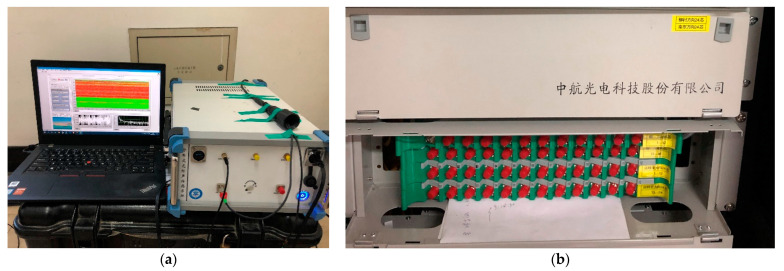
(**a**) The hardware and software of the Φ-OTDR system; (**b**) The sensing fiber-optic cable distribution frame.

**Figure 5 sensors-21-05094-f005:**
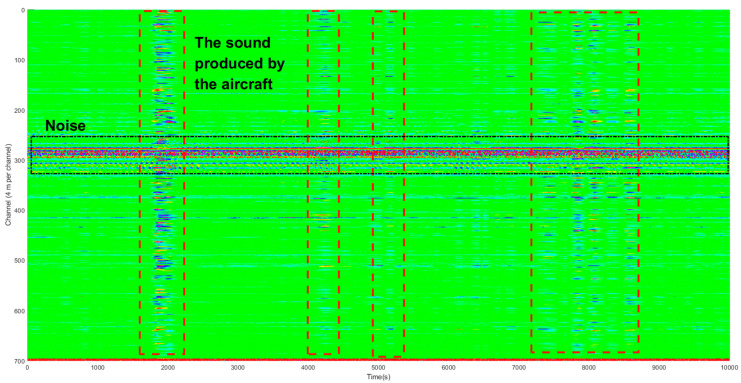
Original signal of an airplane.

**Figure 6 sensors-21-05094-f006:**
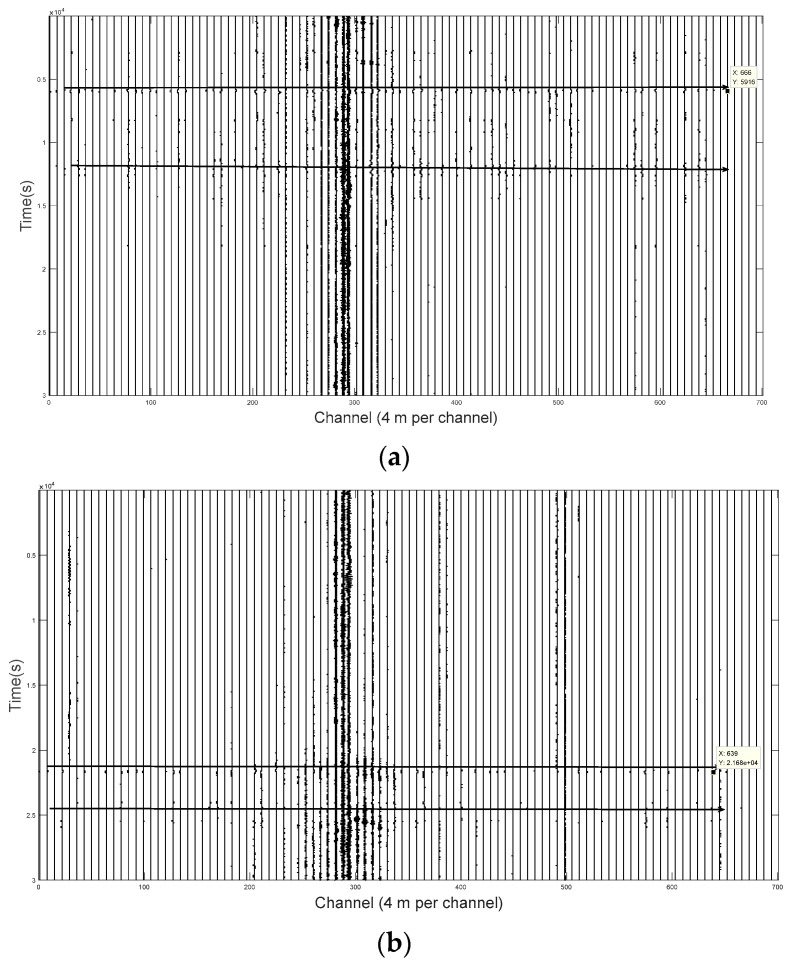
(**a**) Aircraft A vibration signal time delay in different locations of the optic fiber; (**b**) Aircraft B vibration signal time delay in different locations of the optic fiber.

**Figure 7 sensors-21-05094-f007:**
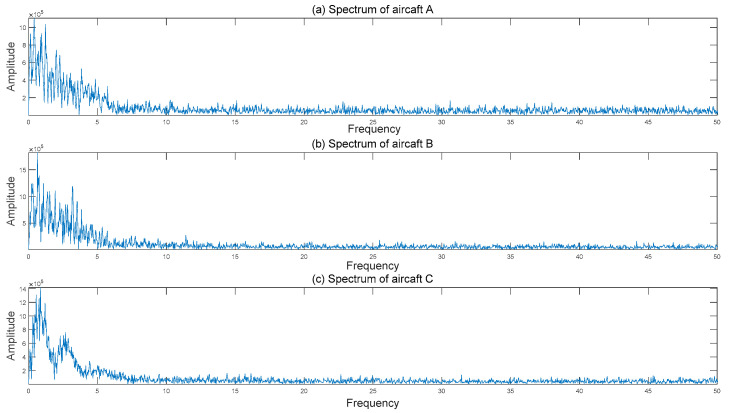
The spectrums of seismic vibration signals from the three aircraft.

**Figure 8 sensors-21-05094-f008:**
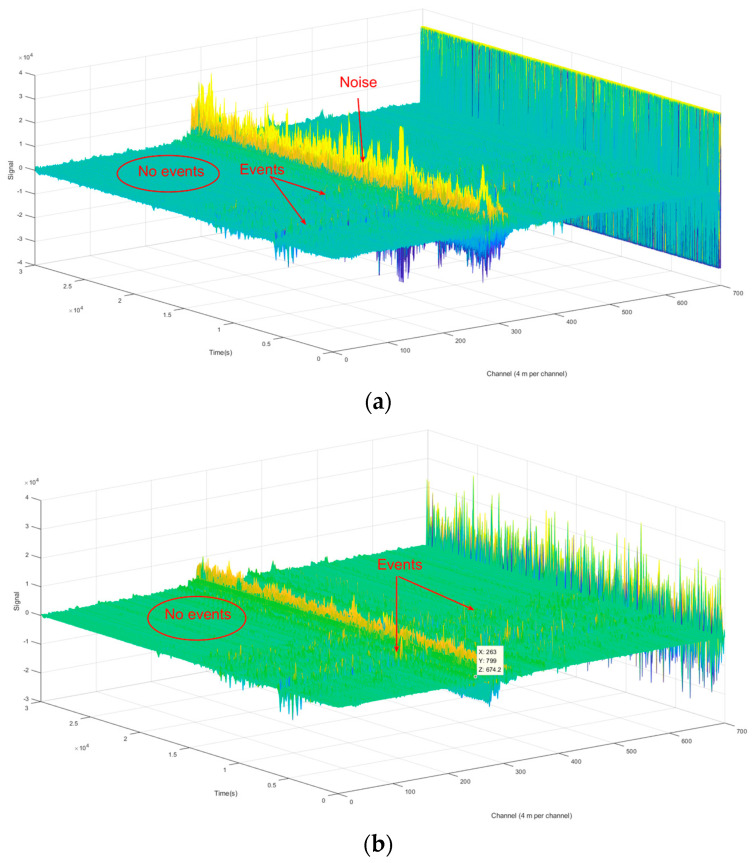
(**a**) Multi-channel signal before filtering; (**b**) Multi-channel signal after filtering; (**c**) Single-channel signal before and after filtering; (**d**) Pass-band curve of the filter.

**Figure 9 sensors-21-05094-f009:**
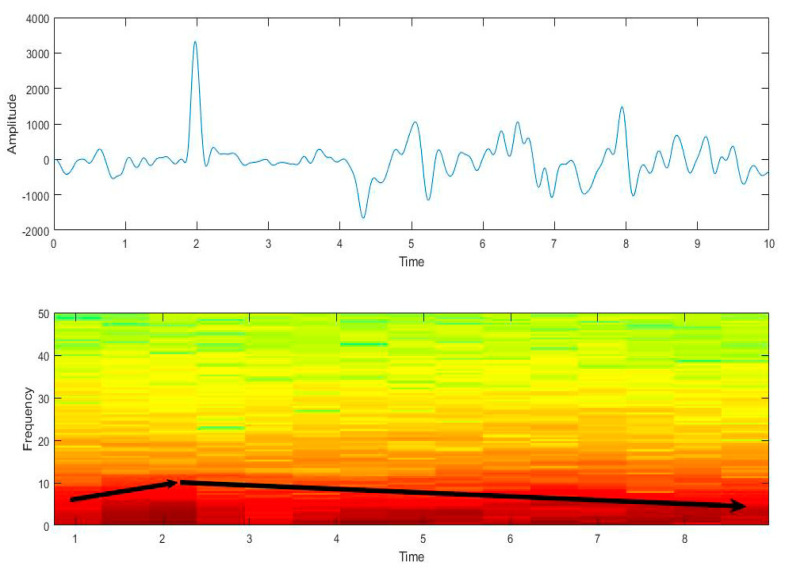
The time-frequency diagram of the airplane vibration signal.

## Data Availability

The data presented in this study are available on request from the corresponding author on reasonable request.

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
