# Peer review of "Aircraft Detection Using Phase-Sensitive Optical-Fiber OTDR"

_sensors, 2021, doi:10.3390/s21155094_

Round 1
Reviewer 1 Report
The topic is interesting. Especially for application in low-frequency noise control that could also be found in automotive
technology or renewable energy production. However, the results are not easy to understand. This is caused by missing information
about the physical quantity that is measured. Furthermore, from a point of measurement technology, quastionas about calibration,
sensor sensitivity, sensing errors are not discussed. However, the application is interesting and, after a revision, of course
intersting for the scientific community.
Line 15: just for my interest: Chebyshev-filter, is the pahse response suitable for the application?
Line 34: delete using?
Line 56: Blank missing before (FOS)
Line 59: FBG not introduced
Line 63: EFPI not introduced
Line 91: the connection to the stress tenso is not celar to me. Also it is not clear, how the author decide between ariborne-noise
and structure-borne noise as well as between airborn-path and structure-born path for sound transmission. It is also not clear, at
least to me, what is the gas? is it the medium aoround the airplaine - the air? i would suggest to explain the meaning between line
89-92 in more detail. Pertherps an illistrating picture could help the redaer.
Figure 2: I can not see the contribution of Figure 2 (a) to the method or the explaination of the method
Line 120-122: what is the difference between spatial resolution (10m) and spatial spamling (4m). the spatial sampling is not a rate
to my unerstanding.
Figure 4: is is possible to explain a little bit more, what are the hardware components? amplifyer? frontend? power supplay?
Figure 5: what is the quantity shown in figure 5? sound pressure? sound intensity? structural intensity? structural velocity in which direction?
any chance to have a magnitude scale?
Figure 6: as figure 5
Line 149: magnitude of time delay?
Figure 7: as figure 5
Figure 8: as figure 5
Reviewer 2 Report
The proposed method is interesting and new. But some sources of noises of acquired signal have to be identified. For example in work: Borecki, M.; Rychlik, A.; Olejnik, A.; Prus, P.; Szmidt, J.; Korwin-Pawlowski, M.L. Application of Wireless Accelerometer Mounted on Wheel Rim for Parked Car Monitoring. Sensors 2020, 20, 6088. https://doi.org/10.3390/s20216088 the parked car rim vibration measurement is related to wheel bolts unscrewing and passing by vehicles. The noise signal of passing by vehicles are analysed for different vehicles and different surface of roads. The observed effect is that when crack is present in road surface the acoustic surface wave is excited with significant amplitude. In case of submitted article airport and landing or plane taking off situation the speed of vehicle may be 200km/s distance between wheels 10m, so approximate vibration frequency is 5.5Hz. Postulated situation and calculated value it is too close value to obtained by authors to be omitted in the description of the experiment. For referring to this situation please give a signals registration when:
- there are not moving planes on airport and there are no flying planes.
- a plane is moving on airport and there is no flying planes over the sensor.
The postulated vibration values in “discussion” section lines 199-200, refers to helicopters “From the existing works regarding acoustic signal detection of airplanes, the frequency range of the signal can be concluded: the frequency of vibration signals generated by aircraft on the runway could range from about 20 Hz to 200 Hz [7-8]” ->
- Eibl, Eva, P, et al. Helicopter location and tracking using seismometer recordings[J]. Geophysical Journal International, 2017. 261
- Haoran, Meng, Yehuda, et al. Characteristics of Airplanes and Helicopters Recorded by a Dense Seismic Array Near Anza 262 California[J]. Journal of Geophysical Research: Solid Earth, 2018.
Helicopter acts in a different way on the ground surface than moving vehicle or airplane. The difference is also in flying physics. Also a physics of introduction of surface acoustic wave is significantly different for airplanes and helicopters so vibration frequencies may differ significantly. Thus please compare it with pointed above publication of moving vehicles as moving on airport airplanes and vehicles are similar.
Minor issues of paper are
- a legend or description of some charts are made with so small font that it is difficult to read.
Please avoid idea shortcuts and jargon that is understanding to authors only, for example:
Line 131 “The data supports this study was collected from October 12 to October 13 in 2020. 700 channels are selected to observe and display the airplane voice signal, the original 132 signal in 10s is shown in Figure 5.” (see above - here is place to make required introduction to presentation of raw signal of experiment including situations that introduce noises.)
Line 149: “As shown in Figure 6, both of the two airplanes only have a short time delay in different parts of the optic fiber.”
Beside pointed issues, that are required to solve, the paper is very interesting.
Round 2
Reviewer 1 Report
I can see s significant improvement oft the paper that is to my knowledge now ready for publication.
Author Response
I'm very glad that the revised paper has met your satisfaction. Thanks again for your time and efforts in helping us improve the paper.